# Shared regulation and functional relevance of local gene co-expression revealed by single cell analysis

Diogo M. Ribeiro [1,2 ✉], Chaymae Ziyani[1,2] & Olivier Delaneau [1,2 ✉]

Most human genes are co-expressed with a nearby gene. Previous studies have revealed this local gene co-expression to be widespread across chromosomes and across dozens of tissues. Yet, so far these studies used bulk RNA-seq, averaging gene expression measurements across millions of cells, thus being unclear if this co-expression stems from transcription events in single cells. Here, we leverage single cell datasets in >85 individuals to identify gene co-expression across cells, unbiased by cell-type heterogeneity and benefiting from the co-occurrence of transcription events in single cells. We discover >3800 co-expressed gene pairs in two human cell types, induced pluripotent stem cells (iPSCs) and lymphoblastoid cell lines (LCLs) and (i) compare single cell to bulk RNA-seq in identifying local gene co-expression, (ii) show that many co-expressed genes – but not the majority – are composed of functionally related genes and (iii) using proteomics data, provide evidence that their co-expression is maintained up to the protein level. Finally, using single cell RNA-sequencing (scRNA-seq) and single cell ATAC-sequencing (scATAC-seq) data for the same single cells, we identify gene-enhancer associations and reveal that >95% of co-expressed gene pairs share regulatory elements. These results elucidate the potential reasons for co-expression in single cell gene regulatory networks and warrant a deeper study of shared regulatory elements, in view of explaining disease comorbidity due to affecting several genes. Our in-depth view of local gene co-expression and regulatory element co-activity advances our understanding of the shared regulatory architecture between genes.

[1] Department of Computational Biology, University of Lausanne, Lausanne, Switzerland. [2] Swiss Institute of Bioinformatics (SIB), Lausanne, Switzerland.
✉email: diogo.ribeiro@unil.ch; olivier.delaneau@unil.ch

The expression of genes is regulated in space and time through the action of various cis-regulatory elements such as promoters, insulators and enhancers[1–4]. These elements play an important role in buffering and fine-tuning gene expression in response to stress conditions, differentiation cues and cell states[5,6]. To achieve a tight control and robust expression level, genes are usually regulated by multiple enhancers, eventually with redundant action[7], as well as multiple target genes[8,9]. Indeed, neighbouring genes frequently exhibit similar behaviour in terms of expression level[10–12]. This local gene co-expression is more pronounced in the immediate vicinity of a gene (e.g., <100 kb) but can occur at longer distances and regardless of the transcriptional orientation or shared functionality[13,14]. In particular, we have previously shown that as many as 59% genes are co-expressed with a nearby gene (within 1 Mb) across 49 GTEx tissues[10]. However, these observations came from studies measuring bulk gene expression in tissues, which entails several limitations. In particular, as these measurements are averages across many sampled cells, the correlation between nearby genes does not necessarily represent co-expression in the same cell. Moreover, tissues contain multiple cell types, which may mask the detection of cell-type-specific co-expression[15,16].

Single-cell analysis has multiple advantages over bulk analysis for addressing the molecular circuitry between gene (co-)expression and regulatory elements by (i) being able to reduce cell-type heterogeneity, or even study one specific cell type[17,18], (ii) producing measurements per cell and thus getting closer in time to the transcription event and (iii) allow to detect co-expression events per individual (i.e., a single genetic background), and thus not affected by linkage disequilibrium (LD). Moreover, with the advent of multimodal single-cell datasets[19,20], chromatin accessibility and gene expression levels can be measured in the same exact cells, which allows exploring the local regulatory elements affecting gene expression at a very high resolution.

Here, we provide an in-depth view of local gene co-expression and regulatory element co-activity using single-cell data in two cell lines (iPSC, LCLs). Namely, we (i) confirm the widespread local co-expression of thousands of gene pairs at the single-cell level, (ii) compare single cell to bulk RNA-seq in identifying local gene co-expression, (iii) explore the co-transcription of co-expressed genes and their maintenance up to the protein level and (iv) identify enhancers involved in local gene co-expression by analysing single-cell RNA-seq and ATAC-seq data performed on the same cells. Our study improves the understanding of the shared regulatory architecture between genes as well as their regulators, which is a prerequisite for understanding their implication in complex traits and disease.

## Results

**Local gene co-expression is widespread in single cells**. To identify locally co-expressed gene pairs (COPs) using single-cell data, we adapted a method developed previously[10] to handle gene expression measurements across thousands of single cells (Fig. 1). Briefly, this method generates genome-wide maps of local gene co-expression from gene expression quantifications by looking at the correlation between genes across cells of the same cell line. For each gene, all other genes in a cis window of 1 Mb around the gene transcription start site (TSS) are tested for having higher than expected expression Pearson correlation. We control for a maximum false discovery rate (FDR) of 5% by comparing the observed correlation to expected correlation values under the null obtained by shuffling expression values 1000 times (see Methods). This approach ensures that differences in the number of nearby genes per region is accounted for. Only autosomal protein-coding

genes were assessed and gene pairs with high cross-mappability, i.e., the extent to which reads from one gene are mapped to the other gene[21], were excluded.

First, we applied this method to gene expression quantifications for undifferentiated induced pluripotent stem cells (iPSC) from 87 individuals from the HipSci consortium[22] for which both single-cell RNA-seq data (Smart-Seq2, 7440 cells in total) and bulk RNA-seq data were available[23,24]. As the single-cell data was obtained in batches which comprised sets of 4 to 6 individuals, to account for potential batch effects, we performed COP identification on a per individual and per experiment basis, as recommended by the data providers[24] (see Methods). Then, for each individual, we obtained the union of COPs stemming from the different experiments (Supplementary Fig. 1). We discovered between 4 and 442 COPs per individual (mean and standard deviation, 113.3 COPs ± 102), the number of COPs identified per individual being strongly correlated with the number of cells available for each individual (Fig. 2a, Spearman $R = 0.88$, $p$-value $= 4.2e^{-29}$), similar to what was observed for the GTEx dataset and tissue sample sizes[10,25]. Across the 87 individuals, we obtained 3877 distinct COPs from single-cell data, out of 254,647 gene pairs tested (Supplementary Table 1 and Supplementary Data 1). Of these, we found 613 COPs present in 2 to 5 individuals and 377 COPs in more than 5 individuals (Fig. 2b). The modest number of COPs found in common between multiple individuals may suggest a high individual and cellular specificity of COPs but may also stem from the lack of power in detecting COPs in individuals with a low number of cells available. Yet, when comparing COP replication rates between experiments of the same individual to experiments from other individuals, we found COP replication to be clearly higher than expected by chance (23.8% versus 2.8%, Wilcoxon test $p$-value $= 3e^{-14}$, Supplementary Fig. 2). Moreover, the 4797 distinct genes composing the 3877 COPs are found widespread across the genome, with between 18.1% and 31.3% of genes per chromosome associated with at least one COP (Supplementary Fig. 3).

In addition to single-cell COPs (scCOPs) mapped for iPSCs across individuals, we also identified 2589 scCOPs from 26,589 cells of a single human lymphoblastoid cell line (LCL, Supplementary Table 1) using the SHARE-seq single-cell dataset[20], a multimodal dataset further explored in the next sections. Data from this dataset includes many more cells but also more sparsity compared to the iPSC dataset (Smart-seq2), which warranted different processing steps such as the removal of genes expressed in <100 cells and binarised expression levels (see Methods). The discovery of several thousand LCL scCOPs in a single individual (compared to hundreds in the iPSC dataset) could be linked to the availability of many thousands of cells and may serve as an example for what may be obtained per individual with larger datasets. Together with the iPSC results, the extensive discovery of COPs using single-cell data demonstrates that local gene co-expression is present for a large proportion of genes and seems to vary across individuals.

**Comparison between single-cell and bulk-derived local gene co-expression**. Next, using bulk RNA-seq data for the same set of 87 individuals, we identified 3705 bulkCOPs by correlating gene expression across individuals (Fig. 1, see Methods). This compares to 3877 distinct scCOPs identified in the same samples, with 313 COPs found in both datasets. This overlap is higher than expected by chance when considering the 239,154 gene pairs tested in both datasets (Fig. 2c, two-sided Fisher's exact test OR = 7.5, $p$-value $= 1.63e^{-148}$). Of note, 311 out of 377 scCOPs replicated across 5 or more individuals were not found with bulk data, yet, many of those are clearly functionally related genes, such as the

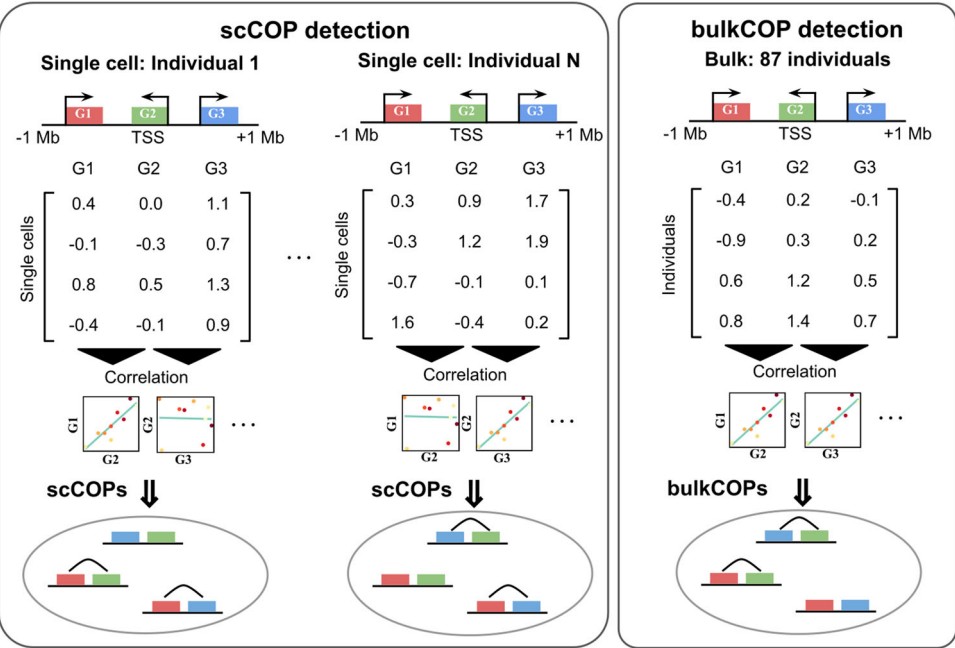

**Fig. 1 Scheme of the single cell and bulk local gene co-expression detection approach across 87 individuals.** Using normalised single-cell data, we identify scCOPs per individual based on measuring the gene expression correlation across all cells of the same individual. Using bulk data (right part of the plot) we identify bulkCOPs by correlating the expression levels of nearby genes across individuals.

histone 1 gene cluster (Supplementary Data 1). Indeed, although the set of COPs differ between single cell and bulk datasets, we found an enrichment for both scCOPs and bulkCOP genes to belong in the same pathway (OR > 1.7, $p$-value < $1.3e^{-27}$, Fig. 2d) and the same protein complex (OR > 11.5, $p$-value < $1.9e^{-22}$). Moreover, both scCOPs and bulkCOPs are enriched in previously identified[10] GTEx COPs, which are conserved across >50% tissues (OR = 6.2 to 15.6, $p$-value < $6.1e^{-42}$). These enrichments greatly increase for the scCOPs replicated across 5 or more individuals (OR = 5.5 to 82.4, $p$-value < $1.4e^{-36}$). We confirmed that the scCOP enrichments are not driven by the COPs that are in common with bulk data (Supplementary Fig. 4). In addition, these enrichments are also replicated on the 2589 scCOPs mapped for LCLs (OR = 1.9 to 5.7, $p$-value < $4.0e^{-10}$, Supplementary Fig. 5). Although highly enriched, the overlap between scCOPs and functionally related gene pairs ranged between 1.3% (same complex) and 16.1% of the scCOPs (same pathway, Fig. 2d), i.e., functionally related genes only represent a minority of all co-expressed gene pairs.

To confirm these results, we analysed single-cell RNA-seq and bulk RNA-seq available for 37 Yoruba individuals of the 1000 Genomes project[26,27]. Notably, we observed very similar results as previous when using this dataset including (i) a correlation between the number of COPs identified per individual and the number of cells available (Spearman $R = 0.66$, $p$-value = $1.1e^{-7}$, Supplementary Fig. 6a), (ii) a similar proportion of COPs being shared across individuals (9% COPs shared across 5 or more individuals, Supplementary Fig. 6b), (iii) matching between the number of COPs identified with bulk RNA-seq (1211 COPs) and single-cell RNA-seq (1155 COPs), with a relatively low but significant overlap between them (two-sided Fisher's exact test OR = 2.38, $p$-value = $5.2e^{-7}$, Supplementary Fig. 6c) and (iv) significant enrichments for COPs to belong to the same gene pathway, protein complex and conserved COPs (Supplementary Fig. 6d).

The fact that scCOPs and bulkCOPs identify different sets of COPs, yet both show strong functional enrichments, suggests that the single-cell COP identification could be complementary to

bulk COP identification in discovering novel and biologically relevant gene co-expression. Out of the 4797 distinct genes present in scCOPs, 1587 genes (33%) are also present in bulkCOPs (Supplementary Fig. 7a). When performing functional enrichments with gProfiler[28] (see Methods) for genes in scCOPs and for genes in bulkCOPs, we observe several enriched biological process terms in common between the two, in particular those related to transcription such as "gene expression" (GO:0010467, adjusted $p$-value < $2.3e^{-9}$) and "cellular metabolic process" (GO:0044237, adjusted $p$-value < $2.2e^{-10}$) (Supplementary Data 2 and Supplementary Fig. 7b, c). However, while the majority of bulkCOP significant enrichments are also found with scCOPs (65 out of 87, Supplementary Data 2 and Supplementary Fig. 7b), scCOPs revealed an additional 547 significant enrichments (Supplementary Data 2 and Supplementary Fig. 7c). These enrichments included terms related to protein translation, such as "translation" (GO:0006412, adjusted $p$-value = $3.9e^{-26}$), "protein-containing complex subunit organisation" (GO:0043933, adjusted $p$-value = $1.0e^{-26}$) and "Ribosome" (KEGG, adjusted $p$-value = $2.3e^{-21}$). Moreover, scCOPs are also highly enriched in "Cell Cycle" (Reactome, adjusted $p$-value = $3.3e^{-11}$) and "Cellular responses to stress" (Reactome, adjusted $p$-value = $5.1e^{-21}$). The expression of cellular stress response genes due to single-cell RNA-seq preparation has previously been reported[29,30]. While this can be problematic in studies of differential gene expression under multiple conditions, here we measure local gene co-expression, which is largely independent of expression levels. Thus, changes in cell states can be seen as an opportunity of discovering novel co-expression events. Indeed, gene expression differences due to cell cycle phases have been previously studied in this dataset[24]. We have thus discovered COPs per cell cycle phase, by first annotating each of the 7440 cells with its most likely cell cycle phase (see Methods) and then separately identifying COPs per individual from cells of each of the G1 (697 cells, 291 COPs), S (3371 cells, 1786 COPs) and G2M phases (3372 cells, 2483 COPs). We again observed that COP identification is dependent on the sample size for each individual-experiment (Supplementary Fig. 8), which likely

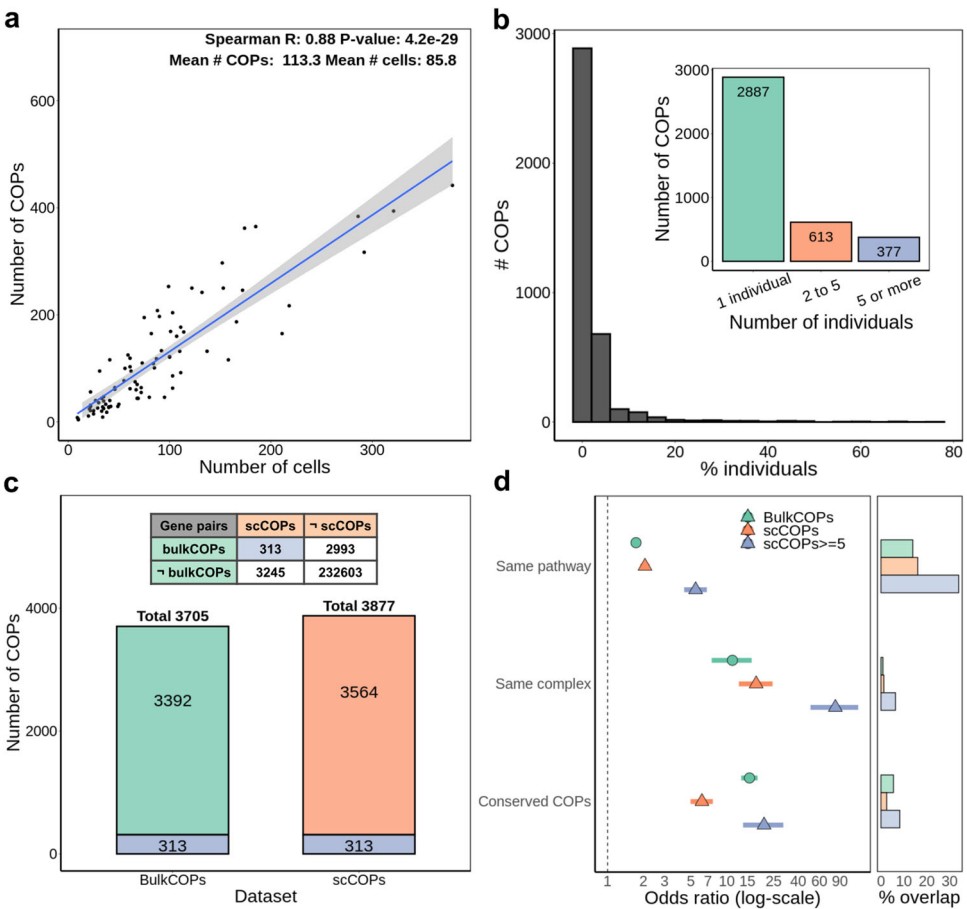

**Fig. 2 Features of single-cell local gene co-expression. a** Number of cells per individual and number of COPs mapped. Fit line corresponds to a linear regression model with 95% confidence intervals; **b** distribution of the percentage of individuals in which COPs are present. The inner plot counts how many COPs in 1, 2 to 5 (exclusive) and 5 or more individuals; **c** total number of COPs detected with bulk data (bulkCOPs) and single-cell data (scCOPs, union across individuals). Numbers in green represent COPs found from both bulk and single-cell data. The contingency table summarises the overlap between scCOPs and bulkCOPs considering the common background of gene pairs tested, which differs slightly between both datasets due to different genes being expressed and detected; **d** one-sided Fisher's exact test odds ratio enrichment (and 95% confidence interval) for the pair of genes in COPs to belong to the same gene pathway, protein complex or in the set of COPs conserved across GTEx tissues. "scCOPs >=5" are a subset of COPs that are found across 5 or more individuals. x-axis is log-scaled, but values shown are before transformation. The right part of the plot denotes the percentage of COPs in each functional annotation.

explains the modest overlap in COPs between phases (e.g., 300 COPs overlapping between S and G2M phases, Supplementary Fig. 9a). However, the enrichment of COPs in genes belonging to the same pathway, protein complex or conserved COPs shows to be very consistent across all phases (Supplementary Fig, 9b). Likewise, many enriched GO terms are present across the 3 cell cycle phases analysed (Supplementary Data 3), including for "Ribosome" (KEGG, adjusted $p$-value $= 6.1e^{-7}$) and "Cellular responses to stress" (Reactome, adjusted $p$-value $= 1.2e^{-6}$), indicating that while COP identification is very much dependent on the available data, the functional enrichments of COP genes are robust. Overall, these results evidence the benefit of using single-cell data to identify COPs in addition to bulk data and the consistent enrichments in pathways and protein complexes reiterate the functional usefulness of local gene co-expression in cells.

**Local gene co-expression is kept up from nascent RNA to protein levels**. In a bid to determine if local gene co-expression not only occurs in the same cells but actually stems from gene transcription at the same time, we analysed gene transcription initiation from a nuclear run-on dataset (GRO-seq) publicly

available for the GM12878 LCL cell line[31]. In practice, we evaluated whether the nascent transcription of both genes in 2589 scCOPs mapped for LCLs is observed, by correlating the number of reads mapping to TSSs of gene pairs (see Methods). Indeed, we found significant read number correlation for COP gene pairs (Spearman $R = 0.22$, $p$-value $= 2.8e^{-4}$, Fig. 3a). To determine if this correlation is higher than expected by the genomic proximity between COP gene TSSs, we produced a control set of 2589 gene pairs that is not co-expressed but closely matches the distance between COPs' TSSs, which we named 'non-COPs' (see Methods). Importantly, in this set of distance-matched non-COP gene pairs we found no read number correlation (Spearman $R = 0.001$, $p$-value $= 0.99$, Supplementary Fig. 10). Given that scCOPs were discovered through their concerted expression in the same cell, this finding suggests that genes in a COP may also be transcribed at the same time.

Next, to address whether the local gene co-expression observed would be relevant at the cellular level, we used bulk proteomics data (MS/MS) from Mirauta et al.[32], which is available for 42 out of the 87 HipSci consortium iPSC cell lines studied here (68 out of 152 individual-experiment combinations). This allowed us to assess whether local gene co-expression correlation can be reproduced as protein intensity correlation (total of 9013 genes

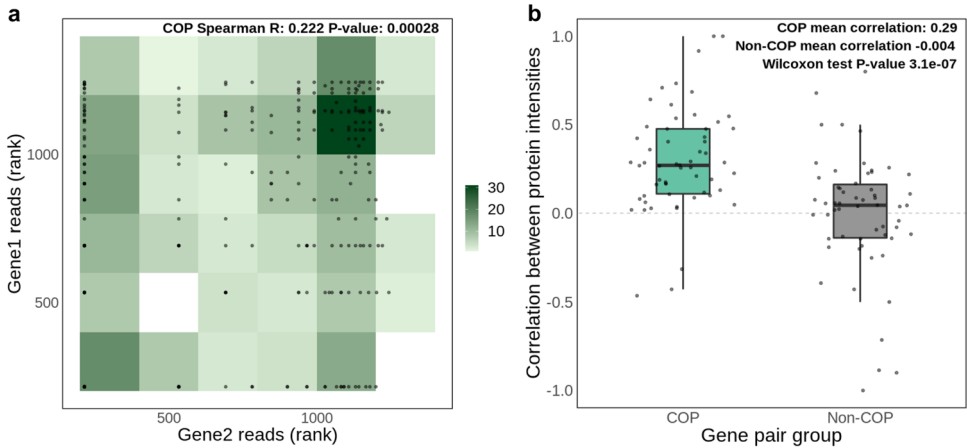

**Fig. 3 GRO-seq and proteomics correlation in scCOPs. a** GRO-seq read correlation for LCL scCOP genes for which data was available ($N = 264$). Reads mapping to the TSS positions of each gene were considered. Gene pairs with missing data in at least one of the genes were excluded. The number of reads across all genes in COPs and non-COPs was ranked prior to plotting. Two genes sharing the same number of reads share the same rank; **b** correlation of protein intensities for each of 58 individual-experiments with iPSC proteomics data. The two-tailed Wilcoxon test refers to the comparison of correlation values between iPSC scCOPs and non-COPs for all 58 individual-experiments. The length of the box corresponds to the interquartile range (IQR) with the centre line corresponding to the median, the upper and lower whiskers represent the largest or lowest value no further than 1.5× IQR from the third and first quartile, respectively.

with protein intensities, see Methods). We found that the 2577 distinct scCOPs discovered in the matching cell lines often display correlated protein intensities, with a mean Spearman correlation of 0.29 across 58 individual-experiment combinations with more than 10 COPs (Fig. 3b). We observed no positive correlation for 2577 distance-matched non-COPs (Spearman $R = -0.004$), indicating that protein intensities correlations are exclusive to COPs (two-sided Wilcoxon test $p$-value $= 3.1e^{-7}$). Moreover, this correlation was not present when shuffling gene pair labels for COPs and non-COPs on each individual-experiment (Supplementary Fig. 11a). Furthermore, when averaging intensities across all the 42 individuals a significant correlation in intensity levels is also observed for scCOPs but not for distance-matched non-COPs (scCOPs Spearman $R = 0.15$ $p$-value $= 4e^{-13}$, non-COPs Spearman $R = -0.05$, $p$-value $= 0.2$, Supplementary Fig. 11b). This was also observed when considering bulkCOPs (COPs Spearman $R = 0.19$ $p$-value $= 2e^{-11}$, non-COPs Spearman $R = 0.04$, $p$-value $= 0.2$, Supplementary Fig. 11c). These results demonstrate that the co-expression of nearby genes often leads to similar protein abundance levels, perhaps unsurprisingly, given the demonstrated functional relatedness between the genes in a COP and their need to ensure co-expression stability.

**Enhancer regulation of local gene co-expression**. To further explore the regulatory mechanisms leading to gene co-expression we utilised publicly available multimodal SHARE-seq data, which simultaneously profiles gene expression (scRNA-seq) and open chromatin (scATAC-seq) on the same single cells[20]. In particular, we used the data available for a human LCL (GM12878), for which 24,844 cells with both scRNA-seq and scATAC-seq are available (see Methods). Focusing on ATAC-seq peaks overlapping known LCL enhancer regions from the EpiMap repository[33], we correlated the activity of enhancers with the expression of nearby genes (+/−1 Mb window of gene TSS, Fig. 4a, see Methods). Out of 350,182 gene-enhancer pairs tested, 32,883 (9.4%) were determined as significant gene-enhancer associations (FDR < 5% from 1000 permutations and Spearman correlation > 0.05, Supplementary Data 4). As expected, significant gene-enhancer associations are more often found at close distances between the gene TSS and the enhancer, yet, in 83% of

associations the gene TSS and enhancers are >100 kb apart (Supplementary Fig. 12). Importantly, we found significant correlations between the gene-enhancer associations and two recent orthogonal datasets of gene-enhancer associations in LCLs: (i) EpiMap[33], based on gene expression and epigenetic modification measurements (Spearman $R = 0.18$, $p$-value $< 2.2e^{-16}$) and (ii) activity-by-contact (ABC) model[9], based on CRISPR perturbations (Spearman $R = 0.06$, $p$-value $3.9e^{-15}$, Supplementary Fig. 13a, b). Of note, these correlation levels are higher than what is obtained when comparing the EpiMap and ABC model orthogonal methods (Spearman $R = 0.04$, $p$-value $= 1.8e^{-6}$, Supplementary Fig. 13c), evidencing their orthogonal discovery of gene-enhancer associations and the lack of a gold standard for evaluating such associations. To further confirm the validity of our gene-enhancer associations, we analysed normalised bulk Hi-C data (5 kb, 10 kb and 25 kb resolution) for LCLs[34]. We find that the correlation level of gene-enhancer associations corresponds to higher Hi-C contact intensities (Spearman $R = 0.13$, $p$-value < 2.2e^{-16}$, Supplementary Fig. 14a). Importantly, this correlation was not observed for distance-matched control regions (see Methods, Supplementary Fig. 14b). Indeed, 22,102 (67.2%) out of the 32,883 significant gene-enhancer associations displayed higher Hi-C contacts than expected by their distance (Supplementary Fig. 14c). These results were reproduced when considering Hi-C resolution of 10 kb or 25 kb (Supplementary Figs. 15 and 16).

Next, using the 32,883 gene-enhancer associations identified, we explored the role of enhancer sharing in local gene co-expression. For this, we first identified enhancers associated with both genes of a gene pair (correlation > 0.05, FDR < 5%, see Methods). Notably, we found that 95.6% of the 2589 COPs identified share at least one enhancer (mean of 6.2 enhancers shared, range: 0 to 26, Fig. 4b, c), significantly more than for 2589 distance-matched non-COPs, where 32.6% shared at least one enhancer (mean of 0.9 enhancers shared, two-sided Fisher's exact test OR = 41.3, $p$-value $< 5e^{-324}$). These results were not driven by a difference in the number of enhancers tested for associations between COPs (21.6 enhancers) and non-COPs (21.5 enhancers, Wilcoxon test $p$-value = 0.6, Supplementary Fig. 17). Remarkably, using the same Hi-C dataset as before, we found that 53.1% of COPs are associated with at least one enhancer displaying high

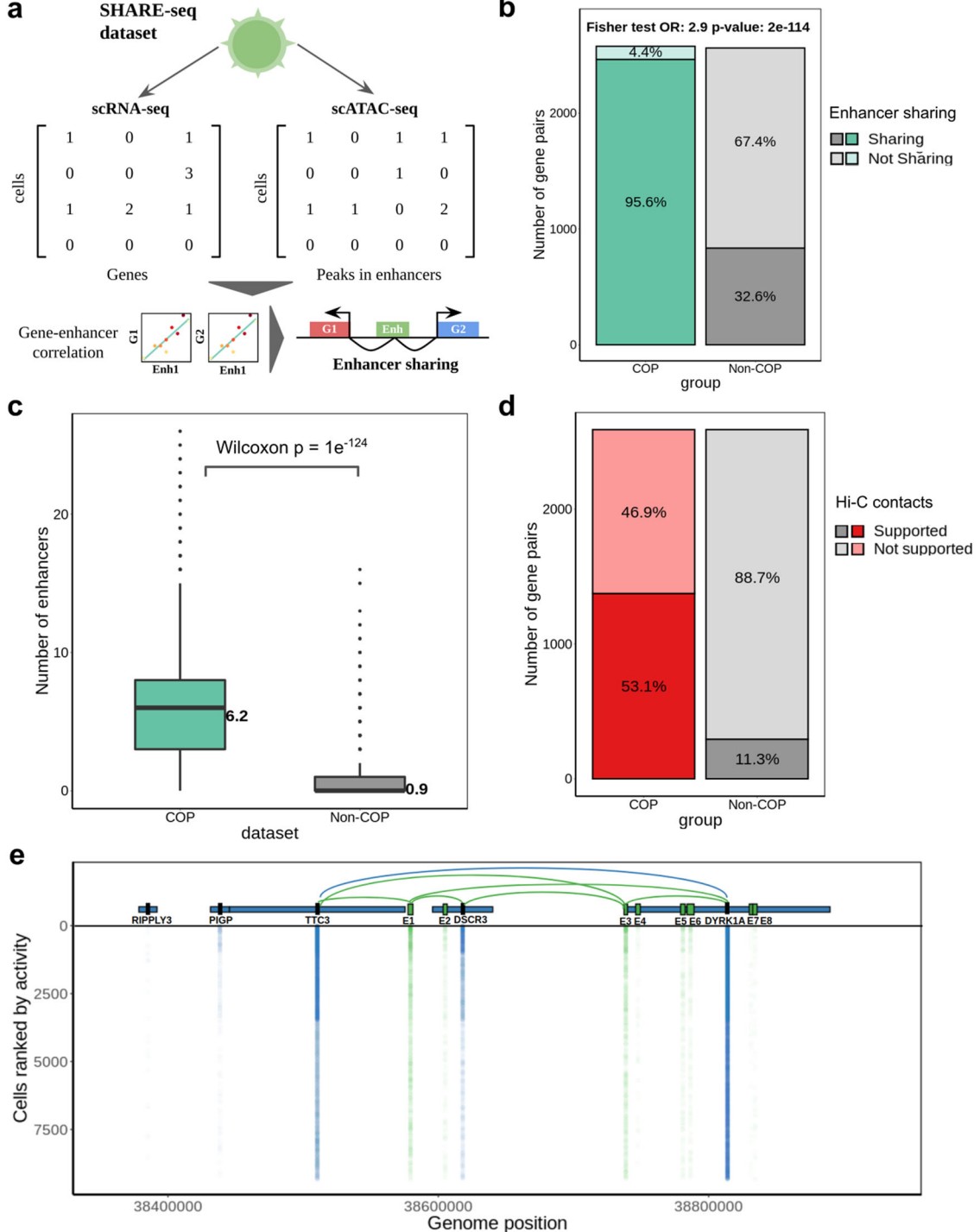

**Fig. 4 Identification of enhancers linked to local gene co-expression. a** SHARE-seq and identification of gene-enhancer associations and enhancer sharing, based on correlating gene expression and chromatin activity on the same cells between nearby genes and enhancer regions ($+/-1$ Mb from gene TSS). Shared enhancers are identified as having a significant association (FDR 5%, Spearman correlation >0.05) with multiple nearby genes; **b** percentage of COPs and non-COPs sharing at least one enhancer, i.e., enhancer significantly associated with both genes in the pair; **c** number of significantly associated enhancers per COP and non-COP. Two-tailed Wilcoxon test was performed between COP and non-COP numbers of enhancers. The length of the box corresponds to the IQR with the centre line corresponding to the median, the upper and lower whiskers represent the largest or lowest value no further than 1.5× IQR from the third and first quartile, respectively; **d** number of COPs and non-COPs with Hi-C support (e.g., both enhancer-gene1 and enhancer-gene2 having Hi-C contact higher than the 75th quantile). Note that non-COPs are less likely to share enhancers and thus a smaller number of gene pairs is liable to have Hi-C support. **e** Overview of the genomic region chr21:38358400-38929000 comprising TTC3 and DYRK1A co-expressed genes, as well as other non-co-expressed genes. The y-axis represents the gene expression (blue) and enhancer region activity (green) across 9341 single cells expressing at least one enhancer or gene. The 8 enhancers found in this region (green models) are denoted as E1 to E8. E1 and E3 are both significantly associated with the TTC3, DSCR3 and DYRK1A genes. The midpoint locations of genes and enhancers are used to draw the y-axis values.

Hi-C contacts to both genes (Fig. 4d, intensities higher than the 75th quantile, see Methods), indicating that these enhancers are likely to interact with both gene's TSSs. For instance, we have found Hi-C support for the co-expression and enhancer sharing of the ST3GAL2 and SF3B3 genes (Supplementary Fig. 18a) and the DRAM1, CCDC53 and NUP37 genes (Supplementary Fig. 18b). Indeed, 76.8% COPs share enhancers with higher Hi-C contacts than distance-matched control regions (Supplementary Fig. 19). Importantly, when considering Hi-C resolutions of 10 kb and 25 kb, similar findings were also observed (Supplementary Figs. 20 and 21). Regulatory element sharing between genes in COPs was also observed when considering all ATAC-seq peaks within +/−100 Kb of each gene TSS (regardless of overlap with known enhancers), with 76.7% COPs sharing peaks compared to 25.1% for non-COPs (Supplementary Fig. 22). Finally, we provide an example visualisation of enhancer sharing between the TTC3 and DYRK1A co-expressed genes, in which two enhancers regulate both genes (Fig. 4e). Overall, the widespread usage of shared regulatory elements shows to be an hallmark of local gene co-expression and may be a key mechanism in ensuring that the transcription of nearby genes is similar and results in stable co-expression.

## Discussion

Previous work described the co-expression of nearby genes through the co-variation of expression levels across individuals for a multitude of tissues[10,12]. However, this work has been achieved with bulk RNA-seq, often in primary tissues, in which the heterogeneity of cell types can confound gene co-expression measurements. Indeed, various studies have shown that the variation in cell-type abundance between samples of the same tissue can affect gene co-expression measurements, i.e., these patterns may reflect the differential expression between cell types of a tissue and even mask robust cell-type-specific co-expression patterns[16,35]. By measuring transcription events in a single cell rather than using gene expression averages, single-cell measurements contain temporality information and may be better suited for the study of local gene co-expression than bulk measurements. Here, we confirmed the widespread presence of local gene co-expression events across the genome in two specific cell types (iPSC and LCL) using single-cell data. Indeed, we discovered more distinct COPs (3877 iPSC COPs) with single-cell data than with bulk data (3705 COPs) for the same set of 87 individuals. Interestingly, only a fraction of these COPs were discovered from both datasets even though the original material came from the same cell lines. Besides technical differences in the library preparation, experimental design, sequencing and processing of bulk and single-cell datasets, obtaining disparate COPs between these approaches is also expected from other points of view. On one hand, when using bulk data to estimate gene co-expression across individuals we leverage interindividual variability in expression levels, which may be controlled by genetic variations. On the other hand, using single-cell data and leveraging inter-cellular variability, the genomes of each cell are the same and gene expression variability may stem from different cell states. These two systems for detecting gene co-expression (perturbation from genetic variants versus perturbation from cell state) are clearly different and thus should be expected to reveal different co-expression events, as observed in other studies[36,37]. Our study confirms this view, by finding a similar global pattern of local gene co-expression between bulk and single-cell COPs (e.g., both are strongly enriched for being in the same pathway), yet each approach finding complementary sets of COPs with different functional enrichments (e.g., scCOP enrichment in cell cycle genes).

A benefit of using single-cell data across multiple individuals is the ability to study the individual-specificity of local gene co-expression. Our results suggest that COPs may be highly individual-specific, however, the limited number of cells per each individual impacts the ability to discover COPs, with only a handful of COPs identified for individuals with less than 20 Smartseq2 cells but with 2589 COPs discovered with >25,000 SHARE-seq cells. Future studies including a higher number of cells per individual as well as more individuals, such as the single-cell eQTLGen consortium[38], will unlock the ability to further explore the interindividual variability of gene co-expression and may even reveal differences between groups of individuals (e.g., based on sex or certain experimental conditions). Indeed, recent studies involving >1 million single cells were able to identify genetic variants associated with gene expression and compare patterns between cases and controls for autoimmune diseases[39,40].

Multimodal single-cell methods performing scRNA-seq and scATAC-seq in the same cells have proven useful in connecting regulatory elements with their target gene expression[20,41,42]. This approach is orthogonal to those using bulk RNA-seq data, such as EpiMap, as the activity of gene expression and regulatory activity is directly observed and correlated in the same cells. Here, we have characterised gene-enhancer links using multimodal data, finding them enriched in Hi-C contacts and the deep resolution of multimodal data allowed us to explore the regulatory circuitry between local gene co-expression and nearby regulatory elements. While the action of multiple regulatory elements on the same gene has often been described[43,44], the action of a regulatory region on multiple genes is seldom explored[20]. Notably, here we found that the vast majority of co-expressed gene pairs (>95%) share at least one regulatory region, much more than expected by chance, indicating this may be a key mechanism for achieving gene co-expression and that the action of enhancers may often be pleiotropic. Further work could unveil not only which regulatory elements are involved in regulating each gene(s) but also provide detail on the actual set of regulatory elements that may work together, by finding evidence for the co-activity of these elements in the same cell.

A current challenge in the field is the inference of gene regulatory networks and determining causality in pathways and gene interactions, with recent studies exploiting single-cell data and deep learning approaches to address this[45,46]. Our finding that (i) local gene co-expression is pervasive and potentially synchronous and (ii) the vast majority of nearby genes share regulatory regions, posits that part of the observed gene co-expression may not reflect gene-gene interactions (e.g., gene1 leading to the expression of gene2), but rather as concomitant events, without a defined directionality between them. As gene co-expression between nearby genes can occur through different molecular cues than co-expression of genes in trans, large-scale analysis, such as deriving pathways and gene regulatory networks, should consider local gene co-expression as a special case warranting specific treatment.

COPs derived from single-cell data are evidence of co-expression in the same cells. For a deeper understanding on the path of gene co-expression, from start to endpoint, we explored datasets of nascent transcription (GRO-seq) and proteomics. We provided evidence that local gene co-expression may stem from the co-transcription of genes. Strikingly, despite the known poor correlation between mRNA and protein levels[47], we observed that gene co-expression leads to matched protein levels. Together with our previous findings regarding (i) functional relatedness of COPs and (ii) the close matching between expression levels and expression variation of co-expressed genes[10], we can stipulate that this consistency in gene and protein levels may derive from the

need for cells to closely match quantities of functionally related proteins. Overall, our results lead us to conceive a model in which (i) genes may be arranged in the genome due to functional relatedness and/or the need to keep a similar level of expression, (ii) nearby co-expressed genes are transcribed at the same time, (iii) the gene co-expression is kept up to the protein level and (iv) this is achieved due to their proximity and sharing of regulatory elements. The potential benefit of such a system could include the minimisation of expression noise and maintenance of appropriate protein complex stoichiometry[48–52]. In agreement with our findings, a strong relationship between chromatin proximity, protein complex interactions and gene co-expression was described in a recent study by Tarbier et al. while measuring genome-wide gene co-variation across hundreds of mouse embryonic stem cells[18].

Our work provides further evidence of the widespread co-expression of nearby genes and explores the regulatory elements involved in their co-expression. With the unravelling of large projects such as the Human Cell Atlas[53], which will contain single-cell RNA-seq complemented with chromatin, protein and spatial information across cell types, as well as studies of context-specific gene expression regulation[36,39], the future promises vast datasets in which to further explore the shared regulatory architecture of gene (co-)expression.

## Methods

**Single-cell datasets used in the study**. We used three datasets of single-cell data in the study. The first dataset was produced by Cuomo et al.[23] from iPSC cell lines from the HipSci consortium[22] and reanalysed in Cuomo et al.[24]. We obtained preprocessed and quality-controlled raw count data from Cuomo et al.[24] (DOI:10.5281/zenodo.4915837), derived from single-cell Smart-Seq2 RNA-seq of undifferentiated iPSCs across 87 individuals, including a total of 7440 cells. We further normalised the gene expression measurements using scran[54] and subsequently rank-transformed the values to match a normal distribution N(0,1). Gene expression counts for an initial 53,958 Ensembl v75 genes were available. From these, genes in non-autosomes (including mitochondrial genes) or the MHC region (chr6:29500000-33600000) were excluded. Gene names were annotated with genomic coordinates (hg19), gene types and Ensembl gene IDs from Gencode v19 and only protein-coding genes were tested for co-expression ($N = 18,943$).

The second single-cell dataset used was obtained from Sarkar et al.[26] (GEO: GSE118723). This dataset contained scRNA-seq in iPSC lines for 54 individuals of the Yoruba population of the 1000 Genomes Project[55]. We obtained preprocessed Smart-Seq2 gene expression quantifications (20,152 Ensembl v75 protein-coding genes, 7584 cells). As done previously, we normalised the gene expression measurements using scran[54] and subsequently rank-transformed the values to match a normal distribution N(0,1). Next, we applied cell and gene filters as suggested by the data providers based on quality controls[26], as well as excluded non-autosomal genes, retaining 9580 genes for analysis.

The third single-cell dataset used in the study was obtained from Ma et al.[20] through GEO (GSE140203). This consisted of preprocessed gene expression counts from the single-cell SHARE-seq method for the GM12878 lymphoblastoid cell line (LCL, GSM4156603, rep3). This dataset included 26,434 expressed genes across 26,589 cells. Cells where <300 or >7500 genes were expressed had been previously removed. As done for the iPSC dataset, we added genomic coordinates (hg19) and Ensembl gene IDs from Gencode v19 and excluded non-protein-coding genes, as well as genes in non-autosomes or in the MHC region. In addition, we excluded genes expressed in less than 100 cells, resulting in a total of 10,821 genes explored for co-expression. Finally, given that >76% of non-zero gene counts were 1 s, the gene expression matrix was binarised (values > 1 became 1, values = 0 remained 0), a common practice, which may aid certain analysis[56] such as gene co-expression.

**Single-cell COP identification**. Single-cell COPs were identified using the previously described method for COP identification in Ribeiro et al.[10]. Briefly, for each gene, we identify all other genes in a cis window of 1 Mb around the gene TSS and compute gene expression correlation (Pearson correlation). We then compare the observed correlation values of each gene/cis-gene pair to expected correlation values under the null derived by shuffling expression values, from which we derive empirical p-values. To exclude potential batch effects, COPs were identified for each individual-experiment combination (total of 152 combinations), as recommended by Cuomo et al.[24]. In each individual-experiment gene expression matrix, genes without any expression variability across the cells (e.g., only zeroes) were excluded. We measured co-expression of all genes within a cis-window of 1 Mb (based on TSS coordinates) and used 1000 permutations to determine observed versus expected empirical p-values, as done before[10]. Positively Pearson correlated

COPs were determined by having a Benjamini–Hochberg (BH) FDR < 5% (the minimum correlation across COPs is 0.248). Of note, due to the rank-transformation step, results between Pearson and Spearman correlation are very similar, Pearson being preferred due to computational speed. Three out 152 individual-experiment combinations were excluded for having a ratio of number of COPs per number of cells above the 75th quartile + interquartile range (IQR) × 3. Moreover, 14,176 gene pairs (out of 268,417 gene pairs within 1 Mb window) with cross-mappability score > 10 were excluded (75mer Exon, 36mer UTR, 2 mismatch, symmetric mean)[21]. Finally, COPs with correlation > 0.99 (i.e., near perfect correlation) were excluded as these are likely to be artefacts. In most analyses, COPs from the different experiments of the same individual were combined together as their union.

Single-cell COPs were also identified for each cell cycle phase. For this, the "CellCycleScoring" function of the Seurat 4.0 R package[57] was used to predict the cell cycle phase for each of the 7440 cells. Then, a matrix was produced with cells of each cell cycle phase (G1, S and G2M) and COP identification for each was performed as described above. Single-cell COPs for the Sarkar et al. dataset[26] were identified for the 37 individuals (one experiment per individual) for which bulk RNA-seq data was also available, using the same parameters and filters as above (1000 permutations, BH FDR 5%, cross-mappability ≤10, correlation ≤0.99, removing outliers above 75th quartile + IQR × 3).

To determine COPs in LCLs from Ma et al.[20] the same parameters were used (1000 permutations, 1 Mb window, cross-mappability score ≤10, correlation ≤ 0.99), with the exception that a minimum Pearson correlation value of 0.2 was used in addition to statistical significance (BH FDR < 5%). Note that for this dataset we used binary data (1 s and 0 s), in which case Pearson and Spearman correlation produces the same exact results. In addition, Mutual Information was found to produce highly similar results. Pearson correlation was preferred due to computation speed.

**Bulk COP identification**. Bulk RNA-seq was produced by the HipSci consortium[22]. Processed and quality-controlled gene expression measurements were obtained from Cuomo et al.[24] for the same set of 87 individuals for which single-cell data was available. As recommended by Cuomo et al. to exclude confounding factors, the first 15 PCA principal components from the expression matrix were regressed out using QTLtools[58]. COPs were identified by correlating expression levels across all 87 individuals, as previously done[10]. A total of 244,341 gene pairs were tested for co-expression. All parameters and post-processing (1 Mb window size, 1000 permutations, 5% FDR, only coding genes, mappability filters) were performed identical as for single-cell COP identification.

In addition, we obtained and processed bulk RNA-seq data for LCLs for 37 Yoruba individuals from the Geuvadis project[27], for which we had single-cell RNA-seq data. Gene expression was quantified for all protein-coding genes annotated in GENCODE v19 (i.e., equivalent to Ensembl v75 used for single-cell data) using QTLtools[58] v1.3 *quan* function with default parameters. As the purpose was to compare with the scRNA-seq dataset, we only retained the 9580 genes which we used in scRNA-seq analysis. We regressed out the first 15 PCA principal components from the expression matrix as before. Parameters and post-processing were performed as above. A total of 65,881 gene pairs were tested for co-expression.

**Creation of control non-COP datasets**. To control for distance effects in local gene pair co-expression, sets of distance-matched non-co-expressed genes (non-COPs) were built. For the 2589 COPs identified in LCLs from Ma et al., we derived non-COPs in the following manner: (1) the pool of tested gene pairs that were not defined as COPs (i.e., Pearson correlation <0.2 and FDR > 5%) were selected ($N = 71,038$), (2) for each of the 2589 COPs, we calculated the absolute distance between the gene TSSs and selected all non-co-expressed gene pairs from the pool, which have an absolute distance ±100 bp of the COP distance value, (3) one of these non-co-expressed gene pairs is randomly selected without replacement and determined as a control non-COPs. Given the large number of initial non-co-expressed gene pairs available, we obtained a non-COP match for each of the 2589 COPs. Sets of non-COPs were identified in this manner also for the Cuomo et al. dataset for each individual-experiment. The only difference being the different cutoff in splitting COPs and non-COPs (based on FDR 5% and not correlation coefficient). For the analysis of all distinct 3877 COPs found across individuals, the pool of non-co-expressed gene pairs used consisted of gene pairs not identified as COPs in any of the individual-experiments. Non-COPs were also identified for bulk COPs in the same manner.

**Enrichment to functionally related gene pairs and annotation terms**. The overrepresentation of COPs as functional-related gene pairs was assessed with one-way Fisher's Exact tests to: (i) genes belonging to the same biological pathway ($N = 24,544$), gathered from KEGG[59] and Reactome[60] through the Ensembl v98 BioMart data mining tool[61] (25 May 2020); (ii) genes belonging to the same human protein complex ($N = 350$), gathered from the CORUM 3.0 database[62] and hu.MAP[63] (20-April-2020). UniprotKB IDs were converted to Ensembl IDs with the Uniprot ID mapping tool[64]. In addition, enrichment tests were performed for a set of COPs conserved across >50% GTEx tissues in which they are expressed

($N = 2441$) from Ribeiro et al.[10]. The background set of gene pairs used included the union of all gene pairs tested for scCOPs and bulkCOPs ($N = 259,834$).

Enrichments to annotation terms for genes in bulkCOPs and scCOPs (union) were performed with the gprofiler2 R package[28,65] (December 2021). For this, a multi-query search was performed for the "GO:BP", "KEGG" and "REACTOME" sources. As the background set, the union of all gene pairs tested for bulkCOPs and scCOPs was used. The results for "GO:BP" were further summarised using REVIGO[66] using the parameters "semantic similarity", "homo sapiens" and "Small (0.5)" return list and plotted with the "TreeMap" R package.

**GRO-seq data and COPs**. Nascent RNA GRO-seq information for the GM12878 LCL was obtained from Core et al.[31] bigWig files. These were converted to bedgraph format using UCSC utils[67] and bedtools intersect was used to combine gene TSS (Gencode v19[68], matching for genomic strand) with the number of GRO-seq reads mapped to the TSS position. A total of 5432 genes had GRO-seq reads in their TSS position. Finally, the number of reads between the two genes in LCL COPs was correlated (Spearman correlation). For a comparison, the same was performed for non-COP gene pairs. Gene pairs with missing data for at least one of the two genes were excluded from the correlation.

**Proteomics data and COPs**. Quantitative proteomic data (Tandem Mass Tag Mass Spectrometry) for 202 iPSC lines derived from 151 individuals of the HipSci consortium samples was collected from Mirauta et al.[32]. This consisted of a processed matrix of protein isoform intensities across individuals, of which 42 are included in the set of 87 individuals assessed here. To measure correlation of protein intensities among COPs and non-COPs, first the protein isoform intensities were converted into 'gene-based intensities' by summing all the protein isoform intensities from each gene. Protein intensities for 9013 genes were obtained in this manner. Next, the intensities for COP gene pairs (and non-COP gene pairs, separately) were correlated (Spearman correlation) per individual-experiment combination, thus producing a correlation value for COPs and a correlation value for non-COPs for each individual-experiment. To ensure the availability of enough data for correlation, only individual-experiment combinations with >10 COPs were considered (a total of 58 out of 68 possible combinations). Gene pairs with missing data for at least one of the two genes were excluded from the correlation. As a control, the genes across the pairs of each individual-experiment COPs/non-COPs were shuffled once.

**Gene-enhancer associations using SHARE-seq**. To identify enhancer regions associated with nearby gene expression, processed and quality-controlled ATAC-seq peaks were retrieved from Ma et al.[20] (GSM4156592, rep3, 507,307 peaks across 67,418 cells). Of these, only 24,844 cells that also had gene expression measurements were kept. GM12878-specific enhancer annotations from the EpiMap repository[33] for hg19 were obtained (18-state chromHMM models). Only regions of genic enhancers (EnhG1, EnhG2) and active enhancers (EnhA1, EnhA2) were considered as enhancer regions. Successive enhancers were merged using bedtools (v2.29.2) *merge* command with default parameters (i.e., only merging "book-ended" features), leading to 33,776 distinct enhancer regions. ATAC-seq peaks were then intersected with these enhancer regions using bedtools intersect with the -F 0.5 parameters, thus requiring that at least 50% of the peak overlaps an enhancer region, resulting in 6,443,451 enhancer-cell combinations (17,765 distinct enhancer regions). Finally, gene expression and open chromatin activity measurements (binarised) were integrated for the same cells and enhancer regions within $+/-1$ Mb of a gene TSS were tested for association with the gene through Spearman correlation. Only protein-coding genes in non-autosomal chromosomes were considered (MHC also excluded). In total, of 350,182 tests were performed. For each test, the expression vector of the gene was shuffled 1000 times and the correlation recalculated. This composes a null distribution from which we derive an empirical p-value for the probability that the observed value is more extreme than the correlations from randomisations. To control for the total number of tests, the Benjamini–Hochberg procedure for FDR was applied on the empirical p-values. Gene-enhancer pairs with correlation coefficient >0.05 and permutation FDR < 5% were identified as significant gene-enhancer associations (32,883 associations). This correlation coefficient was chosen in order to retain a relatively high number of associations while still providing a biological signal. Enhancers significantly associated with both genes of COPs and non-COPs were identified as shared enhancers. The analysis using ATAC-seq peaks (within 100 kb of gene TSS) instead of overlapping with enhancer regions was performed in the same manner as above.

**Comparison to EpiMap and ABC model gene-enhancer associations**. To evaluate the gene-enhancer associations identified here, sets of LCL-specific gene-enhancer maps from the EpiMap repository (links_by_group.lymphoblastoid.tsv.gz)[33] and the activity-by-contact (ABC) model[9] (AllPredictions.AvgHiC.ABC0.015.minus150.ForABCPaperV3.txt) were obtained. The ABC model file was processed to obtain only data for "GM12878-Roadmap" CellType entries and gene names were converted to Ensembl gene IDs using the gprofiler2 R package (gconvert function)[28]. EpiMap and ABC model enhancers were separately intersected with the enhancer regions produced here using bedtools intersect with -wa -wb parameters. Matching gene-enhancer pairs

between datasets ($N = 99,911$ gene-enhancer pairs between EpiMap and SHARE-seq, $N = 19,801$ between ABC model and SHARE-seq) were obtained and the correlation value from the SHARE-seq dataset (no filter) was correlated with EpiMap scores and ABC scores, respectively. As a comparison, EpiMap scores and ABC model scores for matching gene-enhancer pairs were also correlated.

**Hi-C support of gene-enhancer and COP-enhancer associations**. Bulk Hi-C data the GM12878 cell line (LCL) at 5 kb, 10 kb and 25 kb resolution was obtained from Rao et al.[34]. KR normalised (MAPQG0) bins encompassing the TSSs coordinate of gene and midpoint of enhancer regions was obtained through custom Python scripts. Normalised Hi-C contacts were log2-transformed. Using this, gene-enhancer association strength was correlated with Hi-C contacts through Spearman correlation. Gene-enhancer associations with Hi-C contacts above the 75% quantile across all tested gene-enhancer pairs were determined as supported by Hi-C. Missing data (genes or enhancers without Hi-C data) was replaced with 0. As a control, for each gene-enhancer pair, another control pair composed of the gene TSS and an 'enhancer' region on the opposite up- or downstream location in respect to the gene TSS was produced (e.g., if an enhancer is 1000 bp upstream of the gene TSS, the matching control region is 1000 bp downstream of the gene TSS).

**Statistics and reproducibility**. We performed statistical analysis using R programming language, including the scran, Seurat and data.table libraries. Additional software used include Python (including numpy and scipy packages) and gProfiler. COP and gene-enhancer identification included permutation analysis (1000 randomisations), followed by multiple testing correction by applying the Benjamini–Hochberg procedure. In gProfiler, we used the recommended inbuilt 'g:SCS algorithm' for multiple testing corrections.

**Reporting summary**. Further information on research design is available in the Nature Research Reporting Summary linked to this article.

## Data availability
The COPs and enhancer-gene associations produced here are available for download as Supplementary Data 1 to 4, and through the LoCOP public database (https://glcoex.unil.ch/). Source data and code to produce figures is provided in https://github.com/diogomribeiro/sc_cop (DOI: 10.5281/zenodo.6875888). Source data for main figures is also available under Supplementary Data 5. Source data for Fig. 2c is available on Supplementary Data 1. All input data used in this study are available in the public domain. Processed single cell and bulk RNA-seq data from Cuomo et al. is available in a Zenodo repository (DOI:10.5281/zenodo.4915837), whereas Sarkar et al. single-cell data is available through GEO (accession: GSE118723). Bulk RNA-seq of the 1000 Genomes project are available through EBI ArrayExpress (accession: E-GEUV-1). LCL single-cell RNA-seq and ATAC-seq (SHARE-seq) processed data is available through GEO (accession: GSE140203).

## Code availability
The programming code used for data analysis and to produce figures is available under https://github.com/diogomribeiro/sc_cop (https://doi.org/10.5281/zenodo.6875888), together with source data and can be accessed without restrictions.

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

## Acknowledgements

O.D. and D.M.R. have been funded by a Swiss National Science Foundation (SNSF) project grant (PP00P3_176977). D.M.R. has also been funded by the European Union's Horizon 2020 research and innovation programme under the Marie Sklodowska-Curie grant agreement no. 885998. The funders had no role in study design, data collection and analysis, decision to publish, or preparation of the manuscript.

## Author contributions

D.M.R. performed the experiments, analysed the data and wrote the manuscript. C.Z. helped with the gene-enhancer association analysis. O.D. supervised the study and revised the manuscript.

## Competing interests

The authors declare no competing interests.
