## [Peer Review File · Communications Biology]

Reviewers' comments:

Reviewer #1 (Remarks to the Author):

The article 'Shared regulation and functional relevance of local gene co-expression revealed by single cell analysis' by Ribeiro et al. investigates the presence and significance of local co-expressing genes. The authors extend their previous work by translating their method across to single cell RNA-seq (SC RNA-seq) data. The authors were able to show that the local co-expression gene pairs (COPs), which they found in SC RNA-seq data, continued to be correlated downstream in the translational phase and the regulatory phases. The results found by the authors are not completely surprising, as they are in line with our current understanding of the transcriptional mechanism. Nevertheless, the authors performed strong analysis for their findings and the article should be considered for publication. However, prior to this, the reviewer would like the authors to sincerely address the following three major points:

- 1) The 'control non-COP' dataset is very critical to the author's analysis. It is this reviewer's opinion that this section in the methods needs to be expanded and made clearer. The authors state that there are a large number of non-COPs available, hence, a matching non-COP 'was found' for every COP... the authors need to make this step clear. The reviewer can imagine that there can be many corresponding non-COP matches, and choosing one would cause a bias.
- 2) The reviewer understands the need for using correlation based tests for finding gene pairs in their previous work with Bulk-Seq. However, with single cell, we have ample amount of cells, it was shown that more complex measures like Mutual Information (10.1109/BIBM52615.2021.9669880 , 10.1016/j.cels.2017.08.014) are far more suitable. The choice of keeping their correlation based measure needs to be justified and discussed.
- 3) Sadly, the discussion section in the paper is very poor. It is in essence just a restatement of the results. It is not clear to the reviewer what the impact of the authors' findings to the current field of RNA-Seq analysis is. In particular, the field has two very big open questions: Firstly, how to embed scRNA-seq data using Deep Learning (10.1016/j.coisb.2021.05.008, 10.1109/BIBM52615.2021.9669638), and secondly, what are the causal signatures between gene expressions (10.1016/j.patter.2021.100332). The authors need to discuss how their insights into local co-expression patterns can aid or give prior information for furthering SC RNA-seq analysis.

As stated earlier, this is a well written paper and is of high scientific quality. The reviewer would like the three points above to be addressed in revisions, and believes that these points will make the article more helpful and make an impact to the broader RNA-seq analysis community.

Reviewer #2 (Remarks to the Author):

In this manuscript, "Shared regulation and functional relevance of local gene co-expression revealed by single-cell analysis", the Authors show sharing of regulatory elements among the co-expressed genes, specifically in single cells. They claim that more than 95% of co-expressed gene pairs share regulatory elements. I could not locate this analysis with some potential applications on how sharing regulatory elements among the co-expressed genes, specifically in single cells or in bulk data, is helpful. Authors indicate differences in COPs and Non-Cops genes; that is fine, but what does it reveal?

Comments

1. The abstract could have included some potential applications of sharing regulatory elements among the co-expressed genes, specifically in single cells.
2. Authors could provide a separate section for datasets used in this study, including preprocessing of all data sets.

3. Authors should use at least one more data set (comparison in replicates) for Figures 1 and 2.
4. Authors should show the confusion matrix for the relationship between bulk and single cells for COPs and Non-COPs genes and enhancers. Also, conduct some statistics tests might be as fisher or chi-square to show the significance of the confusion matrix.
5. Authors could repeat the study for 10kb, 25kb or more resolution in Hi-C related analysis to reproduce the results.
6. Author can show some Hi-C matrix map (region) and point to the contact frequency where genes share enhancers in COPs and Non-COPs. All Hi-C related analysis points on the Hi-C contact matrix or regions.
7. ATAC-seq peaks, It's difficult to understand how to define the cutoff for peaks; the difference in peaks is calculated as fold change? If not, the same analysis can be redone on fold change. Plot some figures to show the peak intensities for enhancers and genes in COPs and Non-COPs.

Minor comments

1. Abstract: (iPSC and LCL)?
2. Line No 63?
3. Author should provide an explanation or full-form in the first use of short-form or abbreviation.
4. Unique biology?
5. Figure 1:
 - A. Are these results on real data?
 - B. G1, G2, and G3 indicate genes? If yes, what do 0 and 1 values indicate?
 - C. In bulk COP, values are >1? Are these correlation coefficients?
6. Figure 2:
 - A. Y-axis, Log 10 scaled means 10^{1000} is a vast number, total possible pair of genes in close location $\ll 10^{1000}$; explain how?
 - B. The number of COPs increases with the number of cells; one possible reason is the probability of two genes having similar expression increased due to heterogeneity in the gene expression of different cells of the same cell type. The author should mention the significance.
7. Figure 4 Line No. 231:
 - A. Hi-C normalised data?
 - B. Contact between enhancer and gene means higher frequency; is there any cutoff?
 - C. Hi-C data used for this analysis, single or bulk?
 - D. 5KB resolution, means for nearby 1Mb you have analysed for 200 bins or 400 bins?
 - E. ATAC-seq peaks within +/-100Kb? Why 100Kb?

Reply to reviewers

We would like to thank the reviewers for their useful comments and suggestions that helped us improve and clarify the manuscript. Please find below our responses to each of the concerns raised.

Reviewer #1

The article ‘Shared regulation and functional relevance of local gene co-expression revealed by single cell analysis’ by Ribeiro et al. investigates the presence and significance of local co-expressing genes. The authors extend their previous work by translating their method across to single cell RNA-seq (SC RNA-seq) data. The authors were able to show that the local co-expression gene pairs (COPs), which they found in SC RNA-seq data, continued to be correlated downstream in the translational phase and the regulatory phases. The results found by the authors are not completely surprising, as they are in line with our current understanding of the transcriptional mechanism. Nevertheless, the authors performed strong analysis for their findings and the article should be considered for publication. However, prior to this, the reviewer would like the authors to sincerely address the following three major points:

We thank the reviewer for the appreciation of our work. We have addressed all points raised.

1) The ‘control non-COP’ dataset is very critical to the author’s analysis. It is this reviewer’s opinion that this section in the methods needs to be expanded and made clearer. The authors state that there are a large number of non-COPs available, hence, a matching non-COP ‘was found’ for every COP... the authors need to make this step clear. The reviewer can imagine that there can be many corresponding non-COP matches, and choosing one would cause a bias.

We thank the reviewer for the remark and understand the concern with the sampling of non-COPs. As suggested, the procedure used was clarified in the methods section regarding the creation of control non-COP datasets:

“Creation of control non-COP datasets

To control for distance effects in local gene pair co-expression, sets of distance-matched non-co-expressed genes (non-COPs) were built. For the 2589 COPs identified in LCLs from Ma *et al.* 2020, we derived non-COPs in the following manner: (1) the pool of tested gene pairs that were not defined as COPs (i.e. Pearson correlation < 0.2 and FDR>5%) were selected (N = 71,038), (2) for each of the 2589 COPs, we calculated the absolute distance between the gene TSSs and selected all non-co-expressed gene pairs from the pool which have an absolute distance ± 100 bp of the COP distance value, (3) one of these non-co-expressed gene pairs is randomly selected without replacement and determined as a control non-COPs. Given the large number of initial non-co-expressed gene pairs available, we obtained a non-COP match for each of the 2589 COPs. Sets of non-COPs were identified in this manner

also for the Cuomo *et al.* dataset for each individual-experiment. The only differences included the different cutoff in splitting COPs and non-COPs (based FDR 5% and not correlation coefficient). For the analysis of all distinct 3877 COPs found across individuals, the pool of non-co-expressed gene pairs used consisted of gene pairs not identified as COPs in any of the individual-experiments. Non-COPs were also identified for bulk COPs in the same manner.”

Regarding potential biases in the choice of non-COPs, we have had the same concern previously and performed replication analysis using different random samplings of non-COPs for the key findings in the manuscript. We found little variation in the results across different non-COP randomisations, and in all cases the trend between COPs and non-COPs was found consistent. For instance, in 4 randomisations, the proteomics gene pair correlation for non-COPs varied between -0.004 and 0.062 (correlation being 0.29 for COPs, Revision Fig. 1). Likewise, the percentage of non-COPs sharing at least one enhancer ranges from 30.6% to 33.1% (while for COPs this is 95.6%, Revision Fig. 2) and the Hi-C support for gene pairs varied between 10.4% and 11.3% among 4 non-COP randomisations (while this value is 53.1% in COPs, Revision Fig. 3).

Randomisation 1 (in manuscript)

Randomisation 2

Randomisation 3

Randomisation 4

Revision Fig 1. Reproducibility of proteomics support across several non-COP randomisations.

Randomisation 1 (in manuscript)

Randomisation 2

Randomisation 3

Randomisation 4

Revision Fig 2. Reproducibility of enhancer sharing proportions across several non-COP randomisations.

Randomisation 1 (in manuscript)

Randomisation 2

Randomisation 3

Randomisation 4

Revision Fig 3. Reproducibility of Hi-C support across several non-COP randomisations.

2) The reviewer understands the need for using correlation based tests for finding gene pairs in their previous work with Bulk-Seq. However, with single cell, we have ample amount of cells, it was shown that more complex measures like Mutual Information (10.1109/BIBM52615.2021.9669880 , 10.1016/j.cels.2017.08.014) are far more suitable. The choice of keeping their correlation based measure needs to be justified and discussed.

Indeed, Mutual Information and several other complex metrics have been proposed to improve single cell gene co-expression detection. However, we consider that there is no clearly preferable method and correlation approaches continue to be used in recent publications (e.g. PMID: 33116115), including the publications from which our analysis is derived (PMID: 32251272, PMID: 33098772). Importantly, in our approach we perform permutation analysis, and consider a gene pair co-expressed only if their correlation level is higher than expected based on 1000 permutations of their expression levels. In this context, the metric used to assess gene co-expression loses relevance, as the same computation is performed for the permuted expression levels. This is further demonstrated below.

To assess the impact of using MI instead of correlation in our study, we mapped COPs with MI for the LCL SHARE-seq dataset (binary expression values, >20k cells). We found results between MI and correlation to be highly coherent. In fact, correlation and MI across all tested

gene pairs (N=73,626) were highly correlated (Spearman's $\rho > 0.92$, p-value $2.2e^{-16}$, Revision Fig. 3). In terms of impact in COP identification, we had identified 2589 COPs with correlation > 0.2 and FDR $< 5\%$ for the SHARE-seq dataset. For MI, if we pick the 2589 gene pairs with the highest MI (and passing the FDR 5% threshold), all 100% of those COPs were identified with the correlation approach. The mean correlation coefficient of the missing 168 COPs was 0.195, i.e. very close to the 0.2 cutoff used.

Given this, we have modified the following text in the manuscript:

“Note that for this dataset we used binary data (1s and 0s), in which case Pearson and Spearman correlation produces the same exact results. In addition, Mutual Information was found to produce highly similar results. Pearson correlation was preferred due to computation speed”

Revision Fig 3. Comparison between Spearman correlation coefficients and Mutual Information across 73,626 gene pairs for the LCL SHARE-seq dataset.

3) Sadly, the discussion section in the paper is very poor. It is in essence just a restatement of the results. It is not clear to the reviewer what the impact of the authors' findings to the current field of RNA-Seq analysis is. In particular, the field has two very big open question: Firstly, how to embed scRNA-seq data using Deep Learning (10.1016/j.coisb.2021.05.008, 10.1109/BIBM52615.2021.9669638), and secondly, what are the causal signatures between gene expressions (10.1016/j.patter.2021.100332). The authors need to discuss how their insights into local co-expression patterns can aid or give prior information for furthering SC RNA-seq analysis.

We have improved several points of the discussion section, including a new paragraph relating our results with open questions in the field:

“A current challenge in the field is the inference of gene regulatory networks and determining causality in pathways and gene interactions, with recent studies

exploiting single cell data and deep learning approaches to address this (Raharinirina et al. 2021; Yuan and Bar-Joseph 2019). Our finding that (i) local gene co-expression is pervasive and potentially synchronous and (ii) the vast majority of nearby genes share regulatory regions, posits that part of the observed gene co-expression may not reflect gene-gene interactions (e.g. gene1 leading to the expression of gene2), but rather as concomitant events, without a defined directionality between them. As gene co-expression between nearby genes can occur through different molecular cues than co-expression of genes in trans, large-scale analysis, such as deriving pathways and gene regulatory networks, should consider local gene co-expression as a special case warranting a specific treatment.”

Reviewer #2

In this manuscript, "Shared regulation and functional relevance of local gene co-expression revealed by single-cell analysis", the Authors show sharing of regulatory elements among the co-expressed genes, specifically in single cells. They claim that more than 95% of co-expressed gene pairs share regulatory elements. I could not locate this analysis with some potential applications on how sharing regulatory elements among the co-expressed genes, specifically in single cells or in bulk data, is helpful. Authors indicate differences in COPs and Non-Cops genes; that is fine, but what does it reveal?

Our work constitutes fundamental research on the principles of gene co-expression. Until now, it is unclear whether genes are co-expressed due to regulatory factors or simply from the chromatin being open around them (or a combination of both). By identifying enhancers that potentially participate in the co-expression of nearby genes, we provide evidence that enhancers play an active role in gene co-expression. Determining how many and which enhancers are shared helps comprehend, for instance, why some non-coding regulatory mutations affect the expression of only one gene, while others affect multiple genes.

Below we address all the points raised by the reviewer.

1. The abstract could have included some potential applications of sharing regulatory elements among the co-expressed genes, specifically in single cells.

We now mention the following potential applications in the abstract:

“Finally, using scRNA-seq and scATAC-seq data for the same single cells, we identify gene-enhancer associations and reveal that >95% of co-expressed gene pairs share regulatory elements. **These results elucidate the potential reasons for co-expression in single cell gene regulatory networks and warrant a deeper study of shared regulatory elements, in view of explaining disease comorbidity due to affecting several genes.** Our in-depth view of local gene co-expression and regulatory element co-activity advances our understanding of the shared regulatory architecture between genes.”

2. Authors could provide a separate section for datasets used in this study, including preprocessing of all data sets.

We have added a new “Single cell datasets used in the study” section providing a more comprehensive overview of the datasets used and their preprocessing.

“We used two datasets of single cell data in the study. The first dataset was produced by Cuomo et al. 2020²³ from iPSC cell lines from the HipSci consortium²² and reanalysed in Cuomo et al. 2021²⁴. We obtained preprocessed and quality-controlled raw count data from Cuomo et al. 2021²⁴ (DOI:10.5281/zenodo.4915837), derived from single cell Smart-Seq2 RNA-seq of undifferentiated iPSCs across 87 individuals, including a total of 7440 cells. We further normalised the gene expression measurements using scran⁵² and subsequently rank-transformed the values to match a normal distribution $N(0,1)$. Gene expression counts for an initial 53,958 Ensembl v75 genes were available. From these, genes in non-autosomes (including mitochondrial genes) or the MHC region (chr6:29500000-33600000) were excluded. Gene names were annotated with genomic coordinates (hg19), gene types and Ensembl gene IDs from Gencode v19 and only protein-coding genes were tested for co-expression ($N = 18,943$).

The second single cell dataset used in the study was obtained from Ma et al. 2020²⁰ through GEO (GSE140203). This consisted of preprocessed gene expression counts from the single cell SHARE-seq method for the GM12878 lymphoblastoid cell line (LCL, GSM4156603, rep3). This dataset included 26,434 expressed genes across 26,589 cells. Cells where <300 or $>7,500$ genes were expressed had been previously removed. As done for the iPSC dataset, we added genomic coordinates (hg19) and Ensembl gene IDs from Gencode v19 and excluded non-protein-coding genes, as well as genes in non-autosomes or in the MHC region. In addition, we excluded genes expressed in less than 100 cells, resulting in a total of 10,821 genes explored for co-expression. Finally, given that $>76\%$ of non-zero gene counts were 1s, the gene expression matrix was binarised (values >1 became 1, values = 0 remained 0), a common practice which may aid certain analysis⁵⁵ such as gene co-expression.”

3. Authors should use at least one more data set (comparison in replicates) for Figures 1 and 2.

We took the reviewer's suggestion into account and analysed another dataset containing both single cell RNA-seq and bulk RNA-seq across the same multiple individuals and believe this has further strengthened our manuscript. We would like to point out that such datasets are very rare in the literature and their analysis constitutes a major effort, involving extensive processing of several data types across many samples. However, we have found and analysed another dataset with SmartSeq2 single cell RNA-seq (Sarkar et al. 2019 PMID: 31002671) across 54 individuals from the 1000 Genomes project Yoruba population, for which bulk RNA-seq is available for 37 of those individuals (Geuvadis project, Lappalainen et al. 2013, PMID: 24037378). To reproduce Figure 2 findings (single cell COPs and comparison to bulk COPs), we have focused on the 37 individuals with both types of data and processed the data in the same way as for the Cuomo et al. 2021 dataset. The overall numbers of COPs are reduced in the Yoruba dataset, reflecting the lower sample size available (37 individuals) compared to the Cuomo et al. 2021 dataset used in the manuscript (87 individuals) and for this reason we decided to use this dataset to replicate our results rather than for discovery. Note that Figure 1 referenced by the reviewer is a scheme of our main dataset and analysis approach, not results.

We have added the replication results with the Yoruba dataset as a new supplementary figure together with a new paragraph in the results and methods section.

“To confirm these results, we analysed single cell RNA-seq and bulk RNA-seq available for 37 Yoruba individuals of the 1000 Genomes project (Sarkar et al. 2019; Lappalainen et al. 2013). Notably, we observed very similar results as previous when using this dataset including (i) a correlation between the number of COPs identified per individual and the number of cells available (Spearman R = 0.66, p-value = 1.1e-7, Supplementary Figure 6a), (ii) a similar proportion of COPs being shared across individuals (9% COPs shared across 5 or more individuals, Supplementary Figure 6b), (iii) matching between the number of COPs identified with bulk RNA-seq (1211 COPs) and single cell RNA-seq (1155 COPs), with a relatively low but significant overlap between them (Fisher’s exact test OR = 2.38, p-value = 5.2e-7, Supplementary Figure 6c) and (iv) significant enrichments for COPs to belong to the same gene pathway, protein complex and conserved COPs (Supplementary Figure 6d).”

Supplementary Figure 6 Single cell COP and bulk COP discovery in the Sarkar et al. 2019 dataset (a) number of cells per individual and number of COPs mapped. Fit line corresponds to a linear regression model with 95% confidence intervals; (b) distribution of the percentage of individuals in which COPs are present. The inner plot counts how many COPs in 1, 2 to 5 (exclusive) and 5 or more individuals; (c) total number of COPs detected with bulk data (bulkCOPs) and single cell data (scCOPs, union across individuals). Numbers in green represent COPs found from both bulk and single cell data. The contingency table summarises the overlap between scCOPs and bulkCOPs considering the common background of gene pairs tested; (d) Fisher's exact test odds ratio enrichment (and 95% confidence interval) for the pair of genes in COPs to belong to the same gene pathway, protein complex or in the set of COPs conserved across GTEx tissues. "scCOPs \geq 5" are a subset of COPs that are found across 5 or more individuals. X-axis is log-scaled, but values shown are before transformation. The right part of the plot denotes the percentage of COPs in each functional annotation.

4. Authors should show the confusion matrix for the relationship between bulk and single cells for COPs and Non-COPs genes and enhancers. Also, conduct some statistics tests might be as fisher or chi-square to show the significance of the confusion matrix.

We thank the referee for the suggestion of improvement. **We have added a contingency table on Figure 2c** comparing the overlap between bulk and single cell COPs, as well as **performed a Fisher's Exact test** with the following modification in the main text:

"This compares to 3877 **distinct** scCOPs identified in the same samples, with 313 COPs found in both datasets. **This overlap is higher than expected by chance when considering the 239,154 gene pairs tested in both datasets** (Figure 2c, Fisher's exact test OR = 7.5, p-value = $1.63e^{-148}$)"

Updated manuscript Figure 2c.

In addition, we have added a contingency table also on Figure 4b (enhancer sharing by genes). The results of a Fisher's Exact test are also provided.

Updated manuscript Figure 4b.

5. Authors could repeat the study for 10kb, 25kb or more resolution in Hi-C related analysis to reproduce the results.

We repeated the Hi-C related analysis using 10kb and 25kb resolution as suggested by the reviewer to reproduce results. The results with 10kb and 25kb were very similar to those previously shown for 5kb resolution. We have added these results as supplementary figures in the manuscript.

Supplementary Figure. Hi-C support of gene-enhancer associations with 10kb resolution. (a) Hi-C contact intensities per gene-enhancer association correlation; (b) Hi-C contact intensities per gene-enhancer association correlations when considering control

‘enhancer’ regions on the opposite up- or down-stream location in respect to the gene TSS; (c) Hi-C contact intensity difference between real and control regions. A shift of the distribution to the right (above 0) represents higher Hi-C contacts in the real data compared to control. Missing data (genes or enhancers without Hi-C data) was replaced with 0.

Supplementary Figure. Hi-C support of gene-enhancer associations with 25kb resolution. (a) Hi-C contact intensities per gene-enhancer association correlation; (b) Hi-C contact intensities per gene-enhancer association correlations when considering control ‘enhancer’ regions on the opposite up- or down-stream location in respect to the gene TSS; (c) Hi-C contact intensity difference between real and control regions. A shift of the distribution to the right (above 0) represents higher Hi-C contacts in the real data compared to control. Missing data (genes or enhancers without Hi-C data) was replaced with 0.

Supplementary Figure. Hi-C contact intensity difference between COPs and non-COPs and real and control regions, for 10kb resolution. a number of COPs and non-COPs with

Hi-C support (e.g. both enhancer-gene1 and enhancer-gene2 having Hi-C contact higher than the 75th quantile). Note that non-COPs are less likely to share enhancers and thus a smaller number of gene pairs is liable to have Hi-C support. **b** Control regions are the opposite up- or down-stream location in respect to the gene TSS. The mean Hi-C contacts between the enhancer and both genes in a COP were used. A shift of the distribution to the right (above 0) represents higher Hi-C contacts in the real data compared to control. Missing data (genes or enhancers without Hi-C data) was replaced with 0.

Supplementary Figure X Hi-C contact intensity difference between COPs and non-COPs and real and control regions, for 25kb resolution. **a** number of COPs and non-COPs with Hi-C support (e.g. both enhancer-gene1 and enhancer-gene2 having Hi-C contact higher than the 75th quantile). Note that non-COPs are less likely to share enhancers and thus a smaller number of gene pairs is liable to have Hi-C support. **b** Control regions are the opposite up- or down-stream location in respect to the gene TSS. The mean Hi-C contacts between the enhancer and both genes in a COP were used. A shift of the distribution to the right (above 0) represents higher Hi-C contacts in the real data compared to control. Missing data (genes or enhancers without Hi-C data) was replaced with 0.

We have modified the main text accordingly:

“Indeed, 22,102 (67.2%) out of the 32,883 significant gene-enhancer associations displayed higher Hi-C contacts than expected by their distance (Supplementary Figure 13c). **These results were reproduced when considering Hi-C resolution of 10kb or 25kb (Supplementary Figures 14 and 15).**”

“Indeed, 76.8% COPs share enhancers with higher Hi-C contacts than distance-matched control regions (Supplementary Figure 175). **Similar findings were also observed when considering Hi-C resolutions of 10kb and 25kb (Supplementary Figures 18 and 19).**”

6. Author can show some Hi-C matrix map (region) and point to the contact frequency where genes share enhancers in COPs and Non-COPs. All Hi-C related analysis points on the Hi-C contact matrix or regions.

We have added a plot demonstrating example Hi-C contacts between a co-expressed gene pairs and shared enhancers. A reference to this has been added to the main text: “For instance, we have found Hi-C support for the co-expression and enhancer sharing of the ST3GAL2 and SF3B3 genes (Supplementary Figure 18a) and the DRAM1, CCDC53 and NUP37 genes (Supplementary Figure 18b).”

Supplementary Figure 18 Example Hi-C contacts between COPs and shared enhancers (a) region surrounding ST3GAL2 and SF3B3 COP and a shared enhancer (chr16:70510200-70512200); (b) region surrounding DRAM1, CCDC53 and NUP37 co-expressed genes, including a region with multiple shared enhancers (within DRAM1 gene model). The blue models depict genes (black dashes represent transcription start site and direction) and blue curves depicts significant gene co-expression (i.e. COP). The green models depict enhancers and green curves depict significant gene-enhancer correlation (correlation > 0.05, FDR < 5%). In addition to the ST3GAL2 and SF3B3 genes, several other

Hi-C contacts can be observed between genes and enhancer regions, denoting the complexity of gene regulation architecture.

7. ATAC-seq peaks, It's difficult to understand how to define the cutoff for peaks; the difference in peaks is calculated as fold change? If not, the same analysis can be redone on fold change. Plot some figures to show the peak intensities for enhancers and genes in COPs and Non-COPs.

We thank the reviewer for the idea of showing a figure with genes in COPs/non-COPs and associated enhancers. **We have added a new panel in figure 4 (see below).**

Figure 4e: overview of the genomic region chr21:38358400-38929000 comprising TTC3 and DYRK1A co-expressed genes, as well as other non-co-expressed genes. The y-axis represents the gene expression (blue) and enhancer region activity (green) across 9341 single cells expressing at least one enhancer or gene. The 8 enhancers found in this region (green models) are denoted as E1 to E8. E1 and E3 are both significantly associated with the TTC3, DSCR3 and DYRK1A genes. The midpoint locations of genes and enhancers are used to draw the y-axis values.

Regarding fold change and cutoffs for peaks, in this study we are analysing processed single cell ATAC-seq peak data from the SHARE-seq methodology performed in the Ma *et al.* 2020 study, for a single cell line/individual (LCL, GM12878). Since only two copies of chromosomes are available per cell, the output of ATAC-seq in single cells can only be either 1 or 2, although finding ATAC-seq reads in both DNA copies only occurs in <3% of the peaks. This means that calculation of fold changes or the use of peak intensity cutoffs is not possible with this data.

Minor comments

1. Abstract: (iPSC and LCL)?

These acronyms have now been explained.

2. Line No 63?

Thank you for finding this mistake, the sentence has been corrected and now reads “This approach ensures that differences in the number of nearby genes per region is **accounted for**”

3. Author should provide an explanation or full-form in the first use of short-form or abbreviation.

This use of acronyms has been improved across the manuscript.

4. Unique biology?

The corresponding results section has now been renamed “**Comparison between single cell and bulk-derived local gene co-expression**”

5. Figure 1:

A. Are these results on real data?

No, this figure represents a scheme, not real data. The Figure legend now indicates that it is a scheme: “Figure 1 **Scheme** of the single cell and bulk local gene co-expression detection **approach used** across 87 individuals. Using **normalised** single cell data, we identify scCOPs per individual based on measuring the gene expression correlation across all cells of the same individual. Using bulk data (right part of the plot) we identify bulkCOPs by correlating the expression levels of nearby genes across individuals.”

B. G1, G2, and G3 indicate genes? If yes, what do 0 and 1 values indicate?

Yes, G1,2 and 3 indicate genes. 0 and 1 values indicate the gene expression measurements (in the case of single cell these are counts)

C. In bulk COP, values are >1? Are these correlation coefficients?

The values represent gene expression, which are normalised (mean 0, sd 1). Correlation is calculated with those expression values for pairs of genes. **We have clarified the figure by adding “G1”, “G2” and “G3” labels to the gene expression matrix.**

6. Figure 2:

A. Y-axis, Log 10 scaled means 10^{1000} is a vast number, total possible pair of genes in close location $\ll 10^{1000}$; explain how?

We thank the reviewer for spotting this issue provenient from the `scale_y_log10()` function in the R language (ggplot2 library). **We now show the distribution without logarithm scaling and the y-axis label and figure legend were adjusted accordingly.**

B. The number of COPs increases with the number of cells; one possible reason is the probability of two genes having similar expression increased due to heterogeneity in the gene expression of different cells of the same cell type. The author should mention the significance.

We thank the reviewer for this insight. We now mention this in the main text “the number of COPs identified per individual being strongly correlated with the number of cells available for each individual (Figure 2a, Spearman R = 0.88, p-value = 4.2e-29), similar to what was observed for the GTEx dataset and tissue sample sizes^{10,25} **and potentially reflecting gene expression heterogeneity across cells.**”

7. Figure 4 Line No. 231:

A. Hi-C normalised data?

Yes, we normalised the Hi-C data. **This is now mentioned in the main text.**

B. Contact between enhancer and gene means higher frequency; is there any cutoff?

We did not use a cutoff on the Hi-C contact frequencies, as we could directly compare correlation coefficient versus Hi-C contact, without requiring the use of an arbitrary cutoff.

C. Hi-C data used for this analysis, single or bulk?

This is bulk Hi-C data, since single cell Hi-C data is still very limited. **We have added this information in the main text and methods.**

D. 5KB resolution, means for nearby 1Mb you have analysed for 200 bins or 400 bins?

For this analysis we considered the Hi-C bin falling in the TSS position of the gene and the Hi-C bin falling in the midpoint position of the enhancer (enhancers are normally 1 to 5kb), as described in the methods section. I.e. for each gene-enhancer pair, we obtain the Hi-C contact value between two 5kb bins. We were not required to scan the whole region around genes (except for the example Hi-C plots).

E. ATAC-seq peaks within +/-100Kb? Why 100Kb?

The choice of using 100kb (instead of 1Mb) for ATAC-seq peaks had to do limitations on the number of gene-enhancer association tests needed to be performed, as the number of ATAC-seq peaks (507,307 peaks) is >28x higher than the number of enhancers used (17,765 enhancers). We intended to use ATAC-seq peaks only as confirmation of the signal observed for enhancer regions and using 100kb sufficed for this purpose.

REVIEWERS' COMMENTS:

Reviewer #1 (Remarks to the Author):

This reviewer's comments were all addressed.

Reviewer #3 (Remarks to the Author):

The authors have addressed all my comments and improved the manuscript.